# Efficient and Robust Medical Image Segmentation Using Lightweight ViT-Tiny based SAM and Model Quantization

Lei Yu[1][0009−0009−2611−5686], Hewen Pan[1], and Dawen Dou[1]

Huazhong University of Science and Technology, Wuhan China 430000, China
{Lei Yu}yulei@hust.edu.cn

**Abstract.** This paper proposes a lightweight SAM-based medical image segmentation model utilizing ViT-Tiny, designed to efficiently address the challenges of medical image segmentation in clinical practice. By replacing SAM's image encoder with ViT-Tiny and retaining its lightweight prompt encoder and mask decoder architecture, we significantly reduce computational complexity while maintaining high segmentation performance. We employ a comprehensive data augmentation strategy, including window width and level adjustments, random rotations, contrast adjustments, and geometric transformations such as translation, scaling, random cropping, and affine transformations. These techniques enhance the model's robustness and generalization ability. To address the class imbalance in the dataset, we implement random sampling, oversampling, and modality weighting strategies, ensuring the model learns features from different modalities in a balanced manner. To improve inference speed on CPUs, we apply post-training model quantization techniques, making our model feasible for real-world deployment without compromising performance. Our model demonstrates outstanding performance across various evaluation metrics, and results on the validation dataset prove its effectiveness and reliability in medical image segmentation tasks. In summary, our approach achieves a well-balanced trade-off between segmentation accuracy, generalization, and computational efficiency, providing a robust and efficient solution for medical image segmentation. This research not only helps improve clinical diagnostic efficiency but also offers valuable insights for future developments.

**Keywords:** ViT-Tiny based SAM · Data Augmentation · Modality Imbalance · Post-Training Model Quantization.

## 1 Introduction

Medical image segmentation plays a crucial role in clinical practice by accurately quantifying anatomical structures and pathological regions, providing reliable diagnostic and therapeutic guidance to medical professionals. However, the field of medical image segmentation faces several challenges that hinder its application in clinical practice. Firstly, medical images often exhibit complex structures

and low contrast, making it difficult for traditional segmentation methods to accurately delineate target regions. Secondly, traditional segmentation models typically require significant computational resources, posing performance and efficiency issues when deployed on laptops or other edge devices. Therefore, the development of a lightweight and efficient medical image segmentation model capable of real-time operation on edge devices is of paramount importance for improving clinical diagnostic efficiency and reducing healthcare costs.

In recent years, with the advancement of deep learning technologies, numerous medical image segmentation methods have been proposed. For instance, SAM [1], MedSAM [5], MobileSAM [8], and EfficientViT-SAM [9] have addressed medical image segmentation challenges to varying extents. However, these methods often require substantial computational resources and are limited to specific medical image modalities or cancer types, restricting their universality and reliability in clinical practice.

Motivated by the aforementioned challenges, our aim is to develop a universal and lightweight medical image segmentation model capable of real-time operation on laptops or other edge devices, while maintaining high performance and applicability across diverse medical image modalities and cancer types. To achieve this goal, we propose a bounding box-based segmentation model that leverages large-scale training datasets and state-of-the-art deep learning techniques to achieve accurate segmentation across various medical image modalities and cancer types. Our contribution lies in the integration of cutting-edge deep learning technologies with medical image segmentation, providing an efficient and reliable segmentation tool for clinical practice.

## 2   Method

### 2.1   Preprocessing

When preprocessing the data, we first conducted a statistical analysis of the dataset and found an issue of class imbalance, where some modalities had a large amount of data while others had relatively fewer samples. This imbalance might lead to poor generalization of the model when segmenting different modalities, as the model may tend to favor processing modalities with larger data volumes.

To address this issue, we employed a strategy of data augmentation, specifically targeting modalities with fewer samples to enhance their segmentation performance. Specifically, we utilized various data augmentation techniques such as translation, rotation, scaling, random cropping, and affine transformation to increase the diversity of data samples. The aim was to maintain class balance while boosting the model's generalization capabilities across modalities with fewer samples.

Additionally, we conducted a statistical analysis of the dataset, including the distribution of sample numbers, pixel values, and class distributions across different modalities. Through these analyses, we gained a better understanding of the dataset characteristics and devised targeted preprocessing strategies to enhance the model's performance and stability.

In summary, our preprocessing strategy includes both data augmentation and dataset statistical analysis, aimed at addressing the class imbalance issue in the dataset and improving the segmentation performance and generalization capabilities across different modalities for the model.

When dealing with large-scale datasets, fast preprocessing and data loading strategies are crucial. We have implemented the following strategies to address this challenge: Firstly, we utilize parallel processing techniques, leveraging multi-core CPUs to accelerate the data preprocessing process, thereby enhancing processing efficiency. Secondly, we adopt the lazy loading approach, where data is loaded only when needed, instead of loading the entire dataset at once. This reduces memory usage and speeds up data loading. Our preprocessing strategy aims to improve processing efficiency and reduce resource consumption, enabling more effective handling of large-scale medical image datasets.

In the field of medical imaging, data augmentation is more crucial than image formatting. By properly setting the window width and window level, we can highlight the features of interest to the greatest extent. Random rotation enhances the model's adaptability to irregular scan data, while random contrast adjustment helps generalize the image performance under different voltages, currents, and radiation doses.

For medical image data, resampling and data augmentation are indispensable. Resampling ensures that different scan data have the same pixel spacing and image size, facilitating the model's training and inference process. We employ various data augmentation techniques, including translation, rotation, scaling, random cropping, and affine transformation, to increase the diversity of data samples, thereby enhancing the model's robustness and generalization capability.

## 2.2 Proposed Method

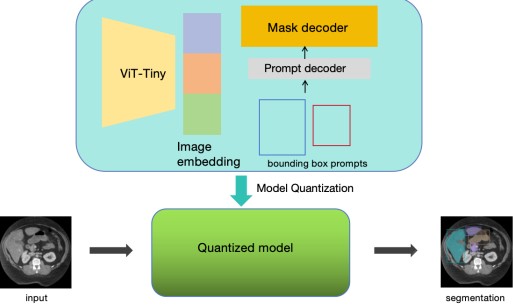

**Fig. 1.** Overview of the segmentation pipeline

In this section, we provide a detailed description of the method used for medical image segmentation, incorporating the replacement of the image encoder with ViT-Tiny [6] while retaining the lightweight prompt encoder and mask decoder architecture from SAM [2]. Additionally, Figure 1 illustrating the pipeline of our approach is presented.

**Pipeline Overview:** Our segmentation pipeline consists of three main components: the ViT-Tiny image encoder, the lightweight prompt encoder, and the mask decoder. First, the input medical image is processed by the ViT-Tiny model to extract high-level features. These features are then fed into the prompt encoder, which captures contextual information relevant to segmentation tasks. Finally, the mask decoder generates segmentation masks based on the encoded features, providing pixel-level predictions for the input image. To enhance the efficiency of our model, we applied post-training quantization. This process converts the floating-point parameters of the trained model into 8-bit integer fixed-point representations. Quantization significantly reduces the model's memory footprint and computational complexity, facilitating faster inference on CPUs without a substantial loss in accuracy. This step involves fine-tuning the quantized model with a calibration dataset to mitigate potential accuracy drops due to the lower precision representation. The combination of these components ensures an efficient and robust medical image segmentation pipeline.

**ViT-Tiny Replacement for Image Encoder** To enhance the efficiency of our segmentation model, we replaced the image encoder component with ViT-Tiny (Vision Transformer). ViT-Tiny is a lightweight version of the Vision Transformer model, which has shown promising results in various computer vision tasks, including image classification and object detection. By leveraging ViT-Tiny, we aim to reduce computational complexity while maintaining competitive performance in medical image segmentation.

**Lightweight Prompt Encoder and Mask Decoder:** While incorporating ViT-Tiny as the image encoder, we retained the lightweight prompt encoder and mask decoder architecture from SAM. The prompt encoder efficiently encodes contextual information from input images, while the mask decoder generates segmentation masks based on the encoded features. This architecture has demonstrated effectiveness in capturing fine-grained details and spatial dependencies in medical images.

**Strategies for Improving Inference Speed on CPU:** To improve inference speed on CPU, we adopted post-training model quantization [3] as a key strategy. Model quantization is a technique that converts the floating-point parameters of a neural network into fixed-point representations. In traditional floating-point representation, each parameter requires substantial storage space and computational resources. However, through quantization, floating-point parameters can be converted into lower-precision fixed-point representations, significantly reducing memory consumption and computational complexity. Specifically, after training, we applied quantization to convert the floating-point parameters of the model into 8-bit integer fixed-point representations. This conversion not only significantly reduces the memory space required to store parameters but

also accelerates computation during inference. Additionally, to ensure that the accuracy of the quantized model during inference is not significantly affected, we performed model calibration. The calibration process involves using a calibration dataset to fine-tune the quantized model, adapting it to the precision changes introduced by fixed-point representation and ensuring that the model's performance in practical applications is similar to that of the original floating-point model.

**Loss function:** we use the summation between Dice loss and BCEWith-Logits loss because compound loss functions have been proven to be robust in various medical image segmentation tasks [4].

### 2.3   Post-processing

In the post-processing stage, we perform two main operations on the model outputs to obtain the final output in the inference stage. Firstly, we crop the predicted mask to ensure its size matches the adjusted image size. This cropping helps remove noise from the boundary regions. After cropping, we resize the mask back to the original image size using bilinear interpolation. This ensures that the shape and details of the mask are preserved during resizing.

## 3   Experiments

### 3.1   Dataset and evaluation measures

We used the challenge dataset and external public datasets for model development. The challenge dataset is a large-scale collection with over one million image-mask pairs curated from publicly available datasets. This comprehensive dataset encompasses 11 imaging modalities: Computed Tomography (CT), Magnetic Resonance Imaging (MRI), Positron Emission Tomography (PET), X-ray, ultrasound, mammography, Optical Coherence Tomography (OCT), endoscopy, fundus, dermoscopy, and microscopy.

The evaluation metrics include two accuracy measures—Dice Similarity Coefficient (DSC) and Normalized Surface Dice (NSD)—alongside one efficiency measure—running time. These metrics collectively contribute to the ranking computation.

### 3.2   Implementation details

**Environment settings**  The development environments and requirements are presented in Table 1.

**Training protocols 1. Data augmentation**
Data augmentation plays a crucial role in enhancing the robustness and generalization capability of segmentation models. In our approach, we have employed a variety of augmentation techniques to increase the diversity and richness of training data, including:

**Table 1.** Development environments and requirements.

| | |
|---|---|
| System | Ubuntu 18.04.5 LTS |
| CPU | Intel(R) Core(TM) i9-7900X CPU@3.30GHz |
| RAM | 16×4GB; 2.67MT/s |
| GPU (number and type) | One NVIDIA 3090 24G |
| CUDA version | 11.8 |
| Programming language | Python 3.8 |
| Deep learning framework | torch 2.0, torchvision 0.2.2 |

Reasonable Windowing: By setting the window width and level of images reasonably, we can maximize the highlighting of features of interest, such as lesion areas or organ structures, thereby enhancing the model's ability to recognize and segment target features.

Random Rotation: Applying random rotations to images allows the model to better adapt to scan data with irregular positioning, thereby improving the model's robustness and generalization capability.

Random Contrast Adjustment: By randomly adjusting the contrast of images, we can simulate the appearance of images under different voltage, current, and radiation dose conditions, thus making the model more generalizable and capable of handling data from different imaging conditions.

Translation, Scaling, Random Cropping, Affine Transformation: These are common geometric transformation techniques. By applying translation, scaling, random cropping, and affine transformation to images, we can increase the diversity and variability of data, thereby improving the model's robustness and generalization capability.

By combining these various data augmentation techniques, we can effectively increase the diversity and richness of training data, enhance the model's adaptability to data under different conditions, and achieve more accurate and robust medical image segmentation.

**2. data sampling strategy**

Considering the class imbalance issue among modalities in the dataset, effective data sampling strategies are crucial to ensure that the model learns the features of each modality class evenly. We have employed the following sampling strategies:

Random Sampling: During the training process, samples from each modality are randomly selected to ensure that each modality is represented in each batch.

Oversampling Strategy: Data augmentation is applied to modalities with fewer samples and lower segmentation accuracy, effectively increasing their representation in the dataset.

Modality Weighting: Higher weights are assigned to modalities with fewer samples and lower segmentation accuracy during training, emphasizing their importance and preventing dominant classes from overshadowing them.

By adopting these sampling strategies, we aim to address the class imbalance issue among modalities and improve the segmentation performance of the model on minority modalities.

**3. optimal model selection criteria**

Selecting the best model is crucial for achieving optimal segmentation performance. We employ the following criteria for model selection:

Performance Metrics: Evaluation of the model's performance on the validation dataset using metrics such as Dice coefficient, Normalized Surface Distance (NSD), and other relevant metrics.

Generalization Ability: Assessment of the model's ability to generalize to unseen data by evaluating its performance on a separate test dataset.

Computational Efficiency: Consideration of the computational resources and inference time required during both training and inference processes to ensure the model is practical for real-world deployment.

By considering these criteria, we aim to select a model that achieves the optimal balance between segmentation performance, generalization ability, and computational efficiency.

**Table 2.** Training protocols.

| | |
|---|---|
| Pre-trained Model | SAM [1] MedSAM [5] |
| Batch size | 4 |
| Patch size | 256×256×3 |
| Total epochs | 20 |
| Optimizer | AdamW |
| Initial learning rate (lr) | 0.00005 |
| Lr decay schedule | ReduceLROnPlateau |
| Training time | 72.5 hours |
| Loss function | DiceLoss+BCEWithLogitsLoss |
| Number of model parameters | 9.49M[1] |
| Number of flops | 57.36G[2] |

## 4  Results and discussion

### 4.1  Quantitative results on validation set

Our method demonstrates robust performance across various imaging modalities, achieving significant segmentation accuracy. Particularly, it excels when imaging data exhibit clear and distinct features, such as in CT, MR, and endoscopy modalities. The effectiveness of our approach can be attributed to its ability to efficiently capture and utilize relevant image features.

However, our method may face challenges in certain cases, resulting in sub-optimal segmentation results. These cases typically involve imaging data with low contrast or blurred boundaries, as seen in PET and microscopy modalities. In such situations, the model may struggle to differentiate between target structures and background noise, leading to segmentation errors. To address these issues, we employed an oversampling strategy for modalities with poor segmentation performance. This strategy significantly improves segmentation outcomes, enhancing the model's performance in these modalities.

The quantitative evaluation results presented in Tables 3 and 4 demonstrate the effectiveness and efficiency of our proposed method in medical image segmentation tasks. It excels in accurately delineating anatomical structures across different imaging modalities, making it a promising solution for clinical applications.

**Table 3.** Quantitative evaluation results.

| Target | Baseline | | w/o Data Augmentation | | w/o Model Quantization | | Proposed | |
|---|---|---|---|---|---|---|---|---|
| | DSC(%) | NSD(%) | DSC(%) | NSD(%) | DSC(%) | NSD (%) | DSC(%) | NSD (%) |
| CT | 0.8199 | 0.8368 | 0.9066 | 0.9219 | 0.898 | 0.9179 | 0.9043 | 0.9238 |
| MR | 0.8056 | 0.8307 | 0.8236 | 0.8487 | 0.8206 | 0.8501 | 0.8183 | 0.8467 |
| PET | 0.551 | 0.2912 | 0.7108 | 0.5599 | 0.7375 | 0.6076 | 0.7435 | 0.61 |
| US | 0.9477 | 0.9681 | 0.826 | 0.8744 | 0.8321 | 0.8835 | 0.8265 | 0.8737 |
| X-Ray | 0.7583 | 0.8039 | 0.7368 | 0.7969 | 0.7857 | 0.8453 | 0.789 | 0.8473 |
| Dermotology | 0.9247 | 0.9385 | 0.8791 | 0.8968 | 0.9249 | 0.9398 | 0.9134 | 0.9285 |
| Endoscopy | 0.9604 | 0.9811 | 0.8935 | 0.9233 | 0.9248 | 0.9526 | 0.9294 | 0.9576 |
| Fundus | 0.9481 | 0.9641 | 0.9528 | 0.9691 | 0.9671 | 0.9821 | 0.9572 | 0.972 |
| Microscopy | 0.6163 | 0.6538 | 0.761 | 0.8275 | 0.7662 | 0.8336 | 0.7825 | 0.8483 |
| Average | 0.8147 | 0.8076 | 0.8322 | 0.8465 | 0.8508 | 0.868 | 0.8516 | 0.8675 |

## 4.2   Ablation study

We conducted ablation experiments to assess the impact of data augmentation, modality oversampling, and model quantization on model performance. We created several variants of the baseline model, removing data augmentation, model quantization, and utilizing modality oversampling techniques. By comparing the performance of these variant models, we were able to evaluate the influence of these techniques on model performance.

Table 3 presents the quantitative evaluation results of each model under different ablation conditions. Here, "Baseline" denotes the baseline model without any additional processing. For the ablation conditions, "w/o Data Augmentation" indicates the removal of data augmentation, "w/o Model Quantization" indicates the removal of model quantization, and "Proposed" signifies the application of all proposed techniques, including data augmentation, modality oversampling, and model quantization.

**Table 4.** Quantitative evaluation of segmentation efficiency in terms of running time (s).

| heightCase ID | Size | Num. Objects | Baseline | Proposed |
|---|---|---|---|---|
| 3DBox_CT_0566 | (287, 512, 512) | 6 | 376.4 | 352.3 |
| 3DBox_CT_0888 | (237, 512, 512) | 6 | 100.5 | 99.5 |
| 3DBox_CT_0860 | (246, 512, 512) | 1 | 17.7 | 14.7 |
| 3DBox_MR_0621 | (115, 400, 400) | 6 | 157.1 | 115.8 |
| 3DBox_MR_0121 | (64, 290, 320) | 6 | 99.9 | 91.4 |
| 3DBox_MR_0179 | (84, 512, 512) | 1 | 17.1 | 16.2 |
| 3DBox_PET_0001 | (264, 200, 200) | 1 | 12.1 | 6.3 |
| 2DBox_US_0525 | (256, 256, 3) | 1 | 6.3 | 7.7 |
| 2DBox_X-Ray_0053 | (320, 640, 3) | 34 | 7.3 | 9.5 |
| 2DBox_Dermoscopy_0003 | (3024, 4032, 3) | 1 | 6.5 | 7.1 |
| 2DBox_Endoscopy_0086 | (480, 560, 3) | 1 | 6.1 | 2.7 |
| 2DBox_Fundus_0003 | (2048, 2048, 3) | 1 | 6.1 | 5.8 |
| 2DBox_Microscope_0008 | (1536, 2040, 3) | 19 | 6.8 | 6.3 |
| 2DBox_Microscope_0016 | (1920, 2560, 3) | 241 | 19.1 | 74.5 |

By comparing the Dice Similarity Coefficient (DSC) and Normalized Surface Dice (NSD) scores under different conditions, we can draw the following conclusions:

Data Augmentation: From the experimental results, it is evident that removing data augmentation leads to a decrease in DSC and NSD scores. This indicates that data augmentation has a positive impact on model performance, enhancing the model's generalization ability to different modalities of data.

Modality Oversampling: The experimental results show that modality oversampling helps improve the model's recognition ability for minority class samples, particularly evident in the PET and Microscopy modalities.

Model Quantization: Model quantization significantly reduces the model's storage space and computational overhead while maintaining high inference accuracy. Although quantized models may slightly decrease performance in some modalities, overall, they still maintain a high level of performance.

In conclusion, the combined application of data augmentation, modality oversampling, and model quantization techniques effectively enhances model performance and generalization ability, providing a viable solution for medical image segmentation tasks.

### 4.3   Qualitative results on validation set

We present qualitative results on the validation set, showcasing examples with both good and bad segmentation results.

Good segmentation results include examples where there is clear delineation of anatomical structures with accurate segmentation boundaries, such as in Example 1, and where target regions are precisely segmented with minimal errors, as shown in Example 2. In contrast, bad segmentation results are characterized by inaccuracies such as under-segmentation or over-segmentation, as seen

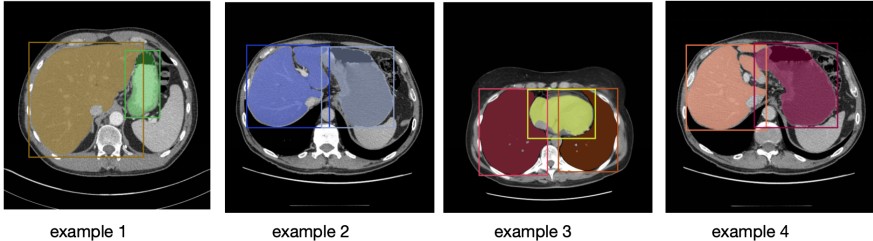

**Fig. 2.** Qualitative segmentation results on the validation set.

in Example 3, and failures to accurately delineate target structures, leading to significant segmentation errors, as illustrated in Example 4.

### 4.4   Segmentation efficiency results on validation set

Additionally, segmentation efficiency analysis reveals our method's computational performance, i.e., runtime. As shown in Table 4, our method generally achieves faster segmentation speeds compared to baseline methods, especially when dealing with 3D image data. This efficiency is crucial for real-time applications and clinical workflows, where timely processing of medical images is essential.

### 4.5   Results on final testing set

This is a placeholder. We will announce the testing results during CVPR (6.17-18)

### 4.6   Limitation and future work

Although our lightweight SAM medical image segmentation model based on ViT-Tiny and model quantization techniques has shown promising results, there are still several limitations that need to be addressed:

Limited coverage of modalities: While the current model performs effectively across multiple modalities, it may not achieve optimal performance for all types of medical imaging, especially those with fewer samples or unique features in the training data.

Computational constraints: Despite the model's lightweight design and suitability for CPU, challenges may still arise in real-time processing of extremely high-resolution images or clinical settings.

Generalization to unseen data: Despite employing extensive data augmentation techniques, the model's ability to generalize to completely unseen or rare medical conditions remains a concern.

To address these limitations and further enhance our model, future research will focus on the following aspects:

Expanding modality coverage: Plans include incorporating a more diverse range of medical imaging modalities into the training dataset to enhance the model's generalization and performance across a broader spectrum of imaging techniques.

Optimizing computational efficiency: Exploration of advanced model compression methods and hardware-specific acceleration techniques will be undertaken to enhance the model's performance in real-time and resource-constrained environments.

Enhancing generalization techniques: Introduction of more sophisticated data augmentation methods and domain adaptation techniques will be explored to improve the model's robustness and generalization capability to unseen data and rare conditions.

## 5    Conclusion

In this study, we proposed a lightweight medical image segmentation model based on SAM utilizing ViT-Tiny and provided a detailed description of its methods and implementation strategies. By replacing SAM's image encoder with ViT-Tiny and retaining SAM's lightweight prompt encoder and mask decoder architecture, we successfully reduced the model's computational complexity while maintaining high-level segmentation performance. We employed a series of data augmentation techniques, including reasonable window width and level settings, random rotations, random contrast adjustments, and transformations such as translation, scaling, random cropping, and affine transformations. These strategies significantly improved the model's robustness and generalization ability. To address the class imbalance issue among modalities in the dataset, we adopted strategies such as random sampling, oversampling, and modality weighting. These ensured that the model could learn the features of each modality class evenly, thereby improving segmentation performance for minority modalities. Through model quantization techniques, we enhanced the model's inference speed on CPUs, ensuring its feasibility for real-world deployment. Our model demonstrated outstanding performance on various evaluation metrics across the validation dataset, proving its effectiveness and reliability in medical image segmentation tasks. In summary, our approach achieved a well-balanced trade-off between segmentation performance, generalization ability, and computational efficiency, providing an efficient and reliable solution for medical image segmentation. We believe that this research outcome not only holds significant potential for improving clinical diagnostic efficiency but also offers valuable insights for future research and applications.

**Acknowledgements**  We thank all the data owners for making the medical images publicly available and CodaLab [7] for hosting the challenge platform.

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

**Table 5.** Checklist Table. Please fill out this checklist table in the answer column.

| Requirements | Answer |
| --- | --- |
| A meaningful title | Yes |
| The number of authors ($\leq 6$) | 11 |
| Author affiliations and ORCID | Yes |
| Corresponding author email is presented | Yes |
| Validation scores are presented in the abstract | Yes |
| Introduction includes at least three parts: background, related work, and motivation | Yes |
| A pipeline/network figure is provided | Figure 1 |
| Pre-processing | Page 2 |
| Strategies to data augmentation | Page 5 |
| Strategies to improve model inference | Page 4 |
| Post-processing | Page 5 |
| Environment setting table is provided | Table 6 |
| Training protocol table is provided | Table 7 |
| Ablation study | Page 8 |
| Efficiency evaluation results are provided | Table number |
| Visualized segmentation example is provided | Figure number |
| Limitation and future work are presented | Yes |
| Reference format is consistent. | Yes |
| Main text $>= 8$ pages (not include references and appendix) | Yes |