# OpenReview forum: "Efficient and Robust Medical Image Segmentation Using Lightweight ViT-Tiny based SAM and Model Quantization"
_thecvf.com/CVPR/2024/Workshop/MedSAMonLaptop — Submitted to CVPR24 MedSAMonLaptop_

### Official Review · Reviewer_E3FL · 2024-06-14
**Too many formatting issues, the author should make a revision**

**Rating:** 4
**Confidence:** 4

**Review:**

The paper proposes a lightweight MedSAM using post-training model quantization techniques and some other techniques. The results show a better performance than the baseline. Here are some advices:

1. Formatting issues.

    1.1 The ORCID for all authors should be included in the paper.

    1.2 The page headers should be rewrited from the template.

    1.3 The last line in Table 1 has an extra blankline.

    1.4 Table 3 exceeds the default layout.

    1.5 In Table 3, it states “DSC(%)” but what follows is not in percentage form.

    1.6 Fig.1 and Fig.2 has wrong font, as it has been requested that the font in   figures should be Times New Roman.

    1.7 Format wrong in Section 3.2, "1. Data augmentation".

2. The description of how the post-training model quantization techniques is implemented should be expanded.

3. Ground truth segmentation results should be added to section 4.3 for better evaluation on the images.

---

### Official Review · Reviewer_4X5V · 2024-06-14
**Lack of method and implementation details, no code provided**

**Rating:** 4
**Confidence:** 4

**Review:**

## Summary
This paper proposes a pipeline to segment multi-modality medical images efficiently on CPUs. This pipeline consists of three main components: a ViT-Tiny based SAM model, extensive data augmentation and sampling, and model quantization. The authors improve the training data diversity and richness through various data augmentation transforms; to address the modality imbalance in the training dataset, they oversample and augment modalities with less training data. However, there is no such information in the paper on which modalities are oversampled and what the augmentation ratio is. Furthermore, for model quantization, it is unclear whether the authors convert a pre-trained model into an 8-bit integer or fine-tune it before quantization. After this quantization, to address the accuracy drop caused by the low precision of an 8-bit integer model, authors fine-tune the quantized model using a calibration dataset, but there is no clue in the paper what this dataset consists of. Overall, this paper lacks methods and implementation details, many statements miss citations or references, and academic writing skill needs to be improved. The GitHub link and code are not provided. The completeness and reproducibility are NOT guaranteed.

## Detailed Comments
**Abstract:**
1-	A good summary of the proposed method. But the DSC or NSD of the validation set were not included.

**Introduction:**
2-	In the first paragraph, “making it difficult for traditional segmentation methods” - what is traditional segmentation methods? There is no reference or citation. The next sentence, “Secondly, traditional segmentation models typically require significant computational resources…”, no reference or citation to “traditional segmentation models” makes this paragraph less coherent and hard to read. The last sentence mentioned “reducing healthcare costs” seems meaningless.

3-	In the second paragraph, it seems the authors would like to introduce medical image segmentation models, but the models they mentioned – SAM, MobileSAM, and EfficientViT-SAM are not proposed to segment medical dat. Although the authors listed some SAM based models, they did not introduce these models. Related works were inadequately demonstrated.

**Methods:**
4-	In the first paragraph, “This imbalance might lead to poor generalization of the model when segmenting different modalities, as the model may tend to favor processing modalities with larger data volumes” This statement needs more demonstration and citation to prove why and how.
5-	The authors conducted a statistical analysis but did not mention what dataset was used, and no tables or figures show the analysis results. The authors included the same data augmentation methods in both the pre-processing session (the second paragraph in session 2.1) and training protocol (session 3.2), was the augmentation done twice or pre-processing and augmentation were duplicated.

6-	“In the field of medical imaging, data augmentation is more crucial than image formatting.” What is the point of mentioning image formatting here? “Random rotation enhances the model’s adaptability to irregular scan data, while random contrast adjustment helps generalize the image performance under different voltages, currents, and radiation doses.” This is a strong argument. Did you conclude this from your experiments? If not, please cite.

7-	"ViT-Tiny Replacement for Image Encoder" needs more explanation on why you need to replace SAM encoder with ViT-Tiny. What is the problem with SAM encoder, is it slower or heavier than ViT-Tiny?

8-	The method figure (Fig 1) does not include all necessary information in the methods and is less informative. What data were used to fine-tune the quantized model?

**Experiments:**
9-	The authors did not mention what and how much data they used for model development, as they said, “external public datasets”.

10-	In data sampling strategy, the author used modality weighting techniques, but they did not list what weights were assigned to what modalities.
11-	In the optimal model selection criteria part, the authors did not introduce what dataset they used to assess model generalizability and what the result is. In the Table 2, the authors listed SAM and MedSAM for training. It is confusing why this sudden appearance of MedSAM, as in the previous content authors only said they used SAM. Importantly, what is the training set and how much data used are unclear.

**Results and discussion:**
12-	In the Table 3, typo for “Dermotology”. Table 3 using % for unit but DSC and NSD scores are out of 1, not 100. Unclear qualitative results, it would be better if having their ground truth to compare. "As shown in Table 4, our method generally achieves faster segmentation speeds compared to baseline methods, especially when dealing with 3D image data." There're cases that proposed model performed worse than the baseline so "generally" might not be appropriate here.

Checklist table
-	Wrong number of authors
-	There is no table 7 and table 8 in the paper.
-	Please add table number for Efficiency evaluation and figure number for Visualized segmentation example.

Please add Abbreviated paper title and First Author Name to headers.

---

### Official Review · Reviewer_V5FK · 2024-06-16

**Rating:** 6
**Confidence:** 5

**Review:**

This paper has sufficient sections and provides detailed information for each section.
However, there are some problems that must be solved.
1. Validation scores are not presented in the abstract.
2. Figure 1 is too small, and the quantized model should be detailed.
3. Please use 'Fig.' as the Figure identifier instead of 'Figure'.
4. No vertical lines in tables.
5. Table 3 exceeds the default layout.
6. The first author and abbreviated paper title are not shown in the header.
Please go through the paper and correct some errors.

---

### Official Review · Reviewer_Wrxq · 2024-06-16
**Needs more innovations over the baseline**

**Rating:** 4
**Confidence:** 5

**Review:**

This work has no major difference than the baseline.

As in https://github.com/bowang-lab/MedSAM/blob/LiteMedSAM/
the baseline is already using the distilled TinyVit model from the original Vit.

The image encoder, prompt encoder and mask decoder are all the same with baseline.

Only improvements are made on data augmentations, data sampling and model quantification, which are marginal in innovations.
Besides, model quantification is not explained in details.

Speed seems to be similar as in baseline, and no average speed is provided.

---

### Decision · Program_Chairs · 2024-10-01

**Decision:**

Major Revision

**Comment:**

Please address the concerns of all reviewers and add testing results. Otherwise, the paper will be rejected in the last round.